# New Solutions in Single-Cell Protein Production from Methane: Construction of Glycogen-Deficient Mutants of *Methylococcus capsulatus* MIR

**Sergey Y. But** [1,2], **Ruslan Z. Suleimanov** [1], **Igor Y. Oshkin** [1], **Olga N. Rozova** [1,2], **Ildar I. Mustakhimov** [1,2], **Nikolai V. Pimenov** [1], **Svetlana N. Dedysh** [1,*] **and Valentina N. Khmelenina** [2]

1    Winogradsky Institute of Microbiology, Research Center of Biotechnology, Russian Academy of Sciences, 119071 Moscow, Russia; sergeybut20063@gmail.com (S.Y.B.); suleimanov-1972@outlook.com (R.Z.S.); ig.owkin@gmail.com (I.Y.O.); rozovaolga1@rambler.ru (O.N.R.); mii80@rambler.ru (I.I.M.); npimenov@mail.ru (N.V.P.)
2    G. K. Skryabin Institute of Biochemistry and Physiology of Microorganisms, Scientific Center for Biological Research, Russian Academy of Sciences, 142290 Pushchino, Russia; khmelenina@rambler.ru
*    Correspondence: dedysh@mail.ru

**Abstract:** The biotechnology of converting methane to single-cell protein (SCP) implies using fast-growing thermotolerant aerobic methanotrophic bacteria. Among the latter, members of the genus *Methylococcus* received significant research attention and are used in operating commercial plants. *Methylococcus capsulatus* MIR is a recently discovered member of this genus with the potential to be used for the purpose of SCP production. Like other *Methylococcus* species, this bacterium stores carbon and energy in the form of glycogen, particularly when grown under nitrogen-limiting conditions. The genome of strain MIR encodes two glycogen synthases, GlgA1 and GlgA2, which are only moderately related to each other. To obtain glycogen-free cell biomass of this methanotroph, glycogen synthase mutants, Δ*glgA1*, Δ*glgA2*, and Δ*glgA1*Δ*glgA2*, were constructed. The mutant lacking both glycogen synthases exhibited a glycogen-deficient phenotype, whereas the intracellular glycogen content was not reduced in strains defective in either GlgA1 or GlgA2, thus suggesting functional redundancy of these enzymes. Inactivation of the *glk* gene encoding glucokinase also resulted in a sharp decrease in glycogen content and accumulation of free glucose in cells. Wild-type strain MIR and the mutant strain Δ*glgA1*Δ*glgA2* were also grown in a bioreactor operated in batch and continuous modes. Cell biomass of Δ*glgA1*Δ*glgA2* mutant obtained during batch cultivation displayed high protein content (71% of dry cell weight (DCW) compared to 54% DCW in wild-type strain) as well as a strong reduction in glycogen content (10.8 mg/g DCW compared to 187.5 mg/g DCW in wild-type strain). The difference in protein and glycogen contents in biomass of these strains produced during continuous cultivation was less pronounced, yet biomass characteristics relevant to SCP production were slightly better for Δ*glgA1*Δ*glgA2* mutant. Genome analysis revealed the presence of *glgA1*-like genes in all methanotrophs of the *Gammaproteobacteria* and *Verrucomicrobia*, while only a very few methanotrophic representatives of the *Alphaproteobacteria* possessed these determinants of glycogen biosynthesis. The *glgA2*-like genes were present only in genomes of gammaproteobacterial methanotrophs with predominantly halo- and thermotolerant phenotypes. The role of glycogen in terms of energy reserve is discussed.

**Keywords:** single-cell protein production; aerobic methanotrophs; *Methylococcus capsulatus*; glycogen; glycogen synthase; glucokinase; genome editing; glycogen-deficient mutants

## 1. Introduction

Microbial protein, also known as single-cell protein (SCP), has been recognized as one of the feasible and sustainable alternatives to animal products in meeting the growing global protein demand [1,2]. A wide variety of fast-growing microorganisms can be

cultivated for SCP production (reviewed in [1]). Among all potential SCP producers, aerobic methanotrophic bacteria that use methane ($CH_4$) as a growth substrate, are of particular importance. Methane (or natural gas) is a relatively cheap and readily available source of carbon. Gas fermenters that produce methane-derived biomass can be built at different scales, with a typical commercial plant size producing 10,000 to 20,000 tons of protein per year [2,3]. In addition, methane-based SCP production technology is highly attractive on the grounds of sustainability and; therefore, is currently on the verge of large-scale commercialization [2,4].

The biotechnology of converting methane to SCP and other value-added products implies using aerobic methanotrophic bacteria [5–9]. A unique trait of these prokaryotes is the use of methane monooxygenase enzymes to catalyze the oxidation of methane to methanol [10–12]. Currently, described aerobic methanotrophs form several coherent phylogenetic clades within *Gamma*- and *Alphaproteobacteria* as well as *Verrucomicrobia*. Among these bacteria, fast-growing thermotolerant members of the genus *Methylococcus* have been extensively studied as producers of methane SCP [13–16]. A number of currently operating commercial plants, such as those owned by UniBio and Calysta Inc., use strains of *Methylococcus capsulatus* as key components of the industrial methane-utilizing microbial consortia.

Despite the long history of fundamental research and industrial applications, some metabolic features of these bacteria remain poorly understood. Like other gammaproteobacterial methanotrophs, *Methylococcus* species store glycogen, especially under imbalanced growth conditions [17–19]. The accumulation of glycogen in cells may result in a loss of protein content, which could affect the biomass quality as a feed source. Despite the long history of using *Methylococcus* species in biotechnology, little is known about the regulation of metabolic routes leading to glycogen biosynthesis in these bacteria. This knowledge may offer new solutions for constructing biotechnologically relevant strains.

Glycogen is a branched homopolysaccharide of α-1,4-linked glucose subunits with α-1,6-linked glucose at the branching points. In bacteria, glycogen synthase (GlgA) and branching enzyme (GlgB) are responsible for glycogen biosynthesis. Glycogen synthesis is also dependent on the activity of ADP glucose pyrophosphorylase (GlgC, EC 2.7.7.27) [20,21]. Anabolic phosphoglucomutase (Pgm) and catabolic glycogen phosphorylase were also demonstrated to play an important role in glycogen accumulation [22–24]. *Enterobacteria*, including *Escherichia coli* and *Salmonella enterica*, were found to store glycogen under limited growth conditions in excess of a carbon source [21]. Glycogen accumulation of up to 30% of dry cell weight (DCW) was reported for the thermotolerant methanotroph *Methylococcus* sp. NCIB 11083 and the halotolerant *Methylotuvimicrobium alcaliphilum* 20Z [17–19]. Nowadays, the metabolic engineering of methanotrophs represents a rapidly evolving field, offering new opportunities for constructing producer strains tailored for use in methane-based biotechnology [25–29].

*Mc. capsulatus* MIR is a fast-growing thermotolerant gammaproteobacterial methanotroph, which was isolated from the activated sludge of a wastewater treatment plant [30]. Unlike many other *Methylococcus* bacteria, strain MIR exhibits the ability to grow on methanol in a concentration range of 0.05 to 3.5% (vol/vol), which provides additional flexibility in biotechnological applications [30]. With this advantage, strain MIR emerges as a promising candidate for studying carbon metabolism in relation to glycogen biosynthesis. The genome of strain MIR encodes both particulate and soluble membrane monooxygenases, MxaFI and XoxF methanol dehydrogenases as well as the ribulose monophosphate pathway (RuMP), the serine pathway and the Calvin–Benson–Bassham (CBB) cycle for carbon assimilation. Strain MIR harbors two glycogen synthase-encoding gene clusters involved in directing carbon excess into glycogen synthesis.

This study was initiated in order to construct glycogen-deficient strains of *Mc. capsulatus* MIR by inactivating the genes encoding glycogen synthases or the gene encoding glucokinase, and evaluating the characteristics of the mutant strains with respect to their potential use for the purposes of SCP production.

## 2. Materials and Methods

### 2.1. Strains Used in This Study

*Mc. capsulatus* MIR was maintained in 200-mL flasks filled with 30 mL mineral medium P of the following composition (g L$^{-1}$): $KNO_3$ (1), $MgSO_4$ (0.2), $CaCl_2$ (0.02), $Na_2$-EDTA (0.005), $FeSO_4 \times 7H_2O$ (0.002), $ZnSO_4 \times 7H_2O$ (0.0001), $MnCl_2 \times 4H_2O$ (0.00003), $CuSO_4 \times 5H_2O$ (0.0001), $CoCl_2 \times 6H_2O$ (0.0002), $NiCl_2 \times 6H_2O$ (0.00002), $Na_2MoO_4$ (0.00003), $H_3BO_3$ (0.0003). To create nitrogen-limited conditions, the concentration of $KNO_3$ in the medium was reduced to 0.29 g L$^{-1}$. If needed, gentamycin (10 µg mL$^{-1}$), 100 µg/mL spectinomycin (100 µg mL$^{-1}$), or kanamycin (50 µg mL$^{-1}$) were added to the medium. After inoculation, the flasks were hermetically closed with silicone rubber septa, 50 mL of methane was injected into the headspace, and the flasks were incubated on a shaker (120 rpm) at 42 °C. *Escherichia coli* strains Top 10 and S-17-1 were maintained at 37 °C in a selective LB broth or agar-solidified medium LB (1.5% Difco agar) [31]. Gentamycin (4 µg mL$^{-1}$) and kanamycin (50 µg mL$^{-1}$) were added if required.

### 2.2. Construction of Mutant Strains Defective in Glycogen Biosynthesis

To knock out the *glgA1* (M3M30_03055; https://mage.genoscope.cns.fr/, accessed on 20 May 2023) and *glgA2* (M3M30_12400) genes, the ~700 bp flanking fragments were amplified from the genomic DNA of strain MIR using primers listed in Table 1. The *glgA1* flanking fragments were cloned into pK18mob vector between *XbaI*, *SphI*, and *SphI*, *HindIII* sites, while *glgA2* flanking fragments were cloned between *EcorI*, *Acc65I* and *XbaI*, *HindIII* sites [32]. The gentamycin or spectinomycin resistance cassettes were cloned between the fragments at the *BamHI* site resulting in the generation of plasmids pk18glgA1-Gm and pk18glgA2-Sp. These vectors were introduced into the cells of strain MIR via conjugation with *E. coli* S17-1. The latter was grown on LB plates, while strain MIR was grown on agar plates with P medium. Cells of these two strains were mixed on the surface of agar plates containing P medium supplemented with 3% (*v/v*) LB, and incubated in a methane-air atmosphere at 37 °C for 2 days. The biomass was then spread on agar plates with medium P containing gentamycin or spectinomycin. The plates were incubated in a methane-air atmosphere at 42 °C until colonies became visible (10–14 days). The clones were selected by resistance to either gentamycin or spectinomycin and sensitivity to kanamycin. The mutant genotype was further confirmed by polymerase chain reaction PCR analysis.

**Table 1.** Primers used in this work.

| Name | Primer (5′-3′) | Target |
|------|----------------|--------|
| **Primers for cloning** | | |
| MIR glg1-up-F | ATATCTAGAACCGGCATTACCATCACGA | Homology region (HR) upstream of *glgA1* gene |
| MIR glg1-up-R | CAAGCATGCAGGAGTGGCGGACGGTGCGA | |
| MIR glg1-dw-F | CAAGCATGCGCTGCTCCATCGCCGAC | HR downstream of *glgA1* gene |
| MIR glg1-dw-R | TCAAAGCTTGCCAAGGAGATCGTGAATTA | |
| MIR glg2-up-F | AGTGAATTCGACCACGACCAGGCCGAGCA | HR upstream of *glgA2* gene |
| MIR glg2-up-R | TCAGGTACCTCCAGTACCGCCGACACCTA | |
| MIR glg2-dw-F | CAAGCATGCCGCTACGACTATTCCTGGA | HR downstream of *glgA2* gene |
| MIR glg2-dw-R | CTTAAGCTTTGAGCCTGGGCGTGTCGTG | |
| 20Z-glg2C-F | AGTGAATTCGGAGGAGACACATGGCAAAGCAAACTACAAC | *glgA2* gene from *Mm. alcaliphilum* 20Z |
| 20Z-glg2C-R | CAAGGATCCTTATTTGTTACGAATATAGTCATAGAT | |
| GLK_MIR-F | TTGGATCCTGATCGGCGGCTATCGCAT | *glk* gene |
| GLK_MIR-R | TTCGATCGTCCTGCGGCTTTTCGTAGT | |
| **Primers for real-time PCR** | | |
| MIR qrpoB-F | CTGGATGCCCTGGTGGAAAT | *rpoB* gene |
| MIR qrpoB-R | ATTCTCCACCATCTCCCCCA | |
| q-pgmMIR-F | CTACGCAAGAAGGTGAAGGTT | *pgm* gene |
| q-pgmMIR-R | TGTTTACGGATTACGCAGGA | |
| q-glgCMIR-F | ACTTCCCGCTGTCCAACTG | *glgC* gene |
| q-glgCMIR-R | CCAAGTTCTGATACACCGCAT | |
| q-glgA1MIR-F | TACCCCTCTGGCTGCTGGA | *glgA1* gene |
| q-glgA1MIR-R | TTCGGCTCGTCGCTCAAAA | |
| q-glgA2MIR-F | GCCAAGAGCCCCCCAGT | *glgA2* gene |

A similar procedure was carried out to generate the Δ*glgA1*Δ*glgA2* strain. The pk18glgA1-Gm vector was introduced to the Δ*glgA2* strain, or alternatively, the pk18glgA2-Sp vector was introduced to the Δ*glgA1*. Colonies were selected on plates with P medium supplemented with gentamycin and spectinomycin. However, as this approach did not allow us to generate mutants, we constructed a vector providing an inducible expression of glycogen synthase. This involved amplifying the *glgA2* gene from the genomic DNA of *Mm. alcaliphilum* 20Z and cloning it between *EcorI* and *BamHI* sites into pCAH01 [5,19]. The resulting vector was introduced into the Δ*glgA1* cells by conjugation as described above. The cells of Δ*glgA1* strain harboring this vector were used for *glgA2* inactivation using a pk18glgA2-Sp plasmid. The mating media and selective media also contained 500 ng/mL anhydrotetracycline. Clones were selected by resistance to gentamycin, spectinomycin, and kanamycin. The genotype was tested by PCR. The obtained mutant strain Δ*glgA1*Δ*glgA2* was cured of plasmid by growth in a kanamycin-free medium, followed by plating on agar media and selecting kanamycin-sensitive colonies. The colonies were additionally screened by PCR for the absence of plasmid using the primers designed for specific amplification of *glgA2* from *Mm. alcaliphilum* 20Z.

To generate the Δ*glk* mutant strain, the complete sequence (1011 bp) of glucokinase-encoding *glk* gene (M3M30_07505) was amplified using GLK_MIR-F and GLK_MIR-R primers (Table 1) and subsequently cloned into the pK18mob plasmid. The 652 bp internal fragment of the *glk* gene was excised using BshTI (AgeI) and PstI restriction endonucleases, and the resulting gap was filled with an 840 bp gentamicin cassette from plasmid p34S-Gm using SacI restriction endonuclease, followed by T4 polymerase treatment. The resulting vector was used for *glk* gene inactivation as described above.

## 2.3. Glycogen and Glucose Extraction and Quantification

Cells of *Mc. capsulatus* MIR and the mutant strains, grown in 200 mL of liquid P medium, were collected using a centrifuge (Beckman Coulter, Pasadena, CA, USA) at $8000\times g$ for 10 min and freeze-dried. Low molecular weight metabolites were extracted from the cells as described earlier [33] and the extracts were used for glucose quantification by ABTS assay [34]. Glycogen was extracted from cells as described elsewhere [19] and quantified using an anthrone reagent (60 mg of anthrone dissolved in 40 mL of 70% sulfuric acid). A total of 100 μL of the sample was mixed with 1 mL of the anthrone reagent and incubated at 100 °C for 15 min. Optical density was measured at 620 nm. Glycogen concentrations were calculated using a calibration curve for glucose standards, with a correction coefficient of 0.9 applied.

## 2.4. Protein Content Assay

A total of 10 mg of dried cell material was suspended in 1 mL of 1 M NaOH and incubated for 5 min at 100 °C. The protein concentration in the resulting solution was measured using the Lowry method [35].

## 2.5. Electron Microscopy

Examination of cell ultrastructure was performed using cell suspensions of wild-type strain MIR and Δ*glgA1*Δ*glgA2* mutant strain collected at the very beginning of the stationary phase. Collected cells were pre-fixed with 2.5% glutaraldehyde in 0.05 M cacodylate buffer (pH 7.3) at 4 °C for 2 h, and post-fixed in in the same buffer containing 1% $OsO_4$ at 4 °C [18]. Fixed and dehydrated cells were embedded in epoxy resin Epon-812 (Sigma-Aldrich, Burlington, MA, USA). Ultrathin sections were placed on copper grids coated with formvar film, post-stained with uranyl acetate for 30 min and with lead citrate for 15 min [36]. The prepared cell specimens were examined using a JEM-1400 transmission electron microscope (JEOL, Tokyo, Japan) under an accelerating voltage of 80 kV.

*2.6. RNA Isolation and Real-Time PCR*

The cultures of *Mc. capsulatus* MIR and Δ*glk* mutant were grown up to $OD_{600}$ ~0.4. 10 mL of stop solution (5% water-saturated phenol in ethanol) was added to the cultures and cells were collected by centrifugation ($3500 \times g$ for 10 min) using Eppendorf MiniSpin plus centrifuge (Eppendorf, Hamburg, Germany). The precipitated cells were washed with DEPC-treated water, centrifuged, and dissolved in 1 mL Trizol® Reagent (Sigma-Aldrich, Burlington, MA, USA). Lysate was stored on ice for 5 min, and then 200 μL chloroform was added. After 15 min incubation on ice, the samples were centrifuged at $10,000 \times g$ for 15 min and water fractions were transferred into clean tubes. After addition of 0.5 mL isopropanol and 1 h incubation at −20 °C, the samples were centrifuged at $1,000 \times g$ for 15 min, and the supernatant was discarded. The precipitate was washed twice with 80% ethanol, dried, and then dissolved in 50 μL of RNase-free water (Evrogen, Moscow, Russia). To remove any remaining DNA, 5U of RNase-free DNaseI (Thermo Fisher Scientific, Waltham, MA, USA) and 50 U of Thermo Scientific RiboLock RNase Inhibitor (Thermo Fisher Scientific, Waltham, MA, USA) were added to each tube, and the latter were incubated for 30 min at 37 °C. A total of 5 μL of 50 mM EDTA was added followed by DNase temperature inactivation (10 min at 65 °C). Finally, the integrity of the RNA samples was estimated by electrophoresis in 1% agarose gel.

cDNA was built using 0.5 μg of total RNA, 20 pmol of the specific primers (Table 1), and an MMLV Reverse Transcriptase kit (Evrogen, Moscow, Russia). Gene expression levels were determined via quantitative real-time polymerase chain reaction using a qPCRmix-HS SYBR Kit (Evrogen, Moscow, Russia) in the DTlite Real-Time PCR system (DNA-technology, Moscow, Russia). The reaction consisted of pre-incubation at 95 °C for 5 min, followed by 40 cycles at 95 °C for 20 s, 58 °C for 15 s, and 72 °C for 20 s. The data were analyzed according to the Pfaffl method [37] using *rpoB* as a reference gene. The reaction without reverse transcription served as a control for the absence of DNA in RNA extracts.

*2.7. Cultivation in a Bioreactor*

Growth characteristics of *Mc. capsulatus* MIR and mutant strains were also assessed by cultivating them in a 1.5 L bioreactor (GPC BIO, Perigny, France) on natural gas. These experiments were performed with the following process parameters: temperature, 42 °C; agitation, 1000 rpm; gas flow rate, 6000 $cm^3$ $h^{-1}$; air flow rate, 18000 $cm^3$ $h^{-1}$. The pH level of 6.3 was controlled by titration with 0.5% $NH_4OH$ solution. The cultivation was performed in a mineral medium of the following composition (mg $L^{-1}$): $(NH_4)_2SO_4$, 200; KCl, 250; $MgSO_4 \times 7H_2O$, 250; $CuSO_4 \times 5H_2O$, 4.4; $ZnSO_4 \times 7H_2O$, 1.8; $MnSO_4 \times 5H_2O$, 1.2; $FeSO_4 \times 7H_2O$, 4.2; $CoSO_4 \times 6H_2O$, 0.58; $Na_2MoO_4 \times 2H_2O$, 0.7; $H_3BO_3$, 0.6; $NiSO_4 \times 6H_2O$, 0.5, and 0.5 mL/L 85% $H_3PO_4$. A freshly grown seed culture of the examined methanotroph was used to inoculate the bioreactor at the initial $OD_{600}$ 0.3. Every two hours, an aliquot of the cell suspension was taken from the bioreactor to determine the $OD_{600}$ value on a Spectroquant Prove 300 spectrophotometer (Merck, Darmstadt, Germany). Batch cultivation was carried out until $OD_{600}$ of 17–19 was reached, after which the bioreactor was switched to a continuous cultivation with a dilution rate of ~0.25 $h^{-1}$. To determine cell dry weight (DCW) and protein content, cells were collected (centrifugation at $14,000 \times g$ for 7 min), frozen at −76 °C overnight, and subsequently lyophilized at −70 °C. Protein concentration in lyophilized biomass was determined by the Kjeldahl method using a Kjeldahl analyzer (FOSS, Stockholm, Sweden). Ammonia concentrations in culture aliquots were determined using a commercial kit (MedEcoTest, Moscow, Russia).

## 3. Results

*3.1. Identification of Glycogen Synthase Genes in the Genome of Mc. capsulatus MIR*

To generate a glycogen-deficient strain, we searched the genome of *Mc. capsulatus* MIR for sequences resembling known glycogen synthases. Two putative glycogen synthase genes, *glgA1* (M3M30_03055) and *glgA2* (M3M30_12400), were identified using the GlgA sequence from *E. coli* K12 as the query in a blast search [38]. Both putative glycogen

synthases belong to the GT1 glycosyl transferase family and are only moderately related to each other (23% of amino acid sequence identity) (Figure 1). The GlgA1 from strain MIR and the corresponding enzyme from *E. coli* K12 display 46% sequence identity.

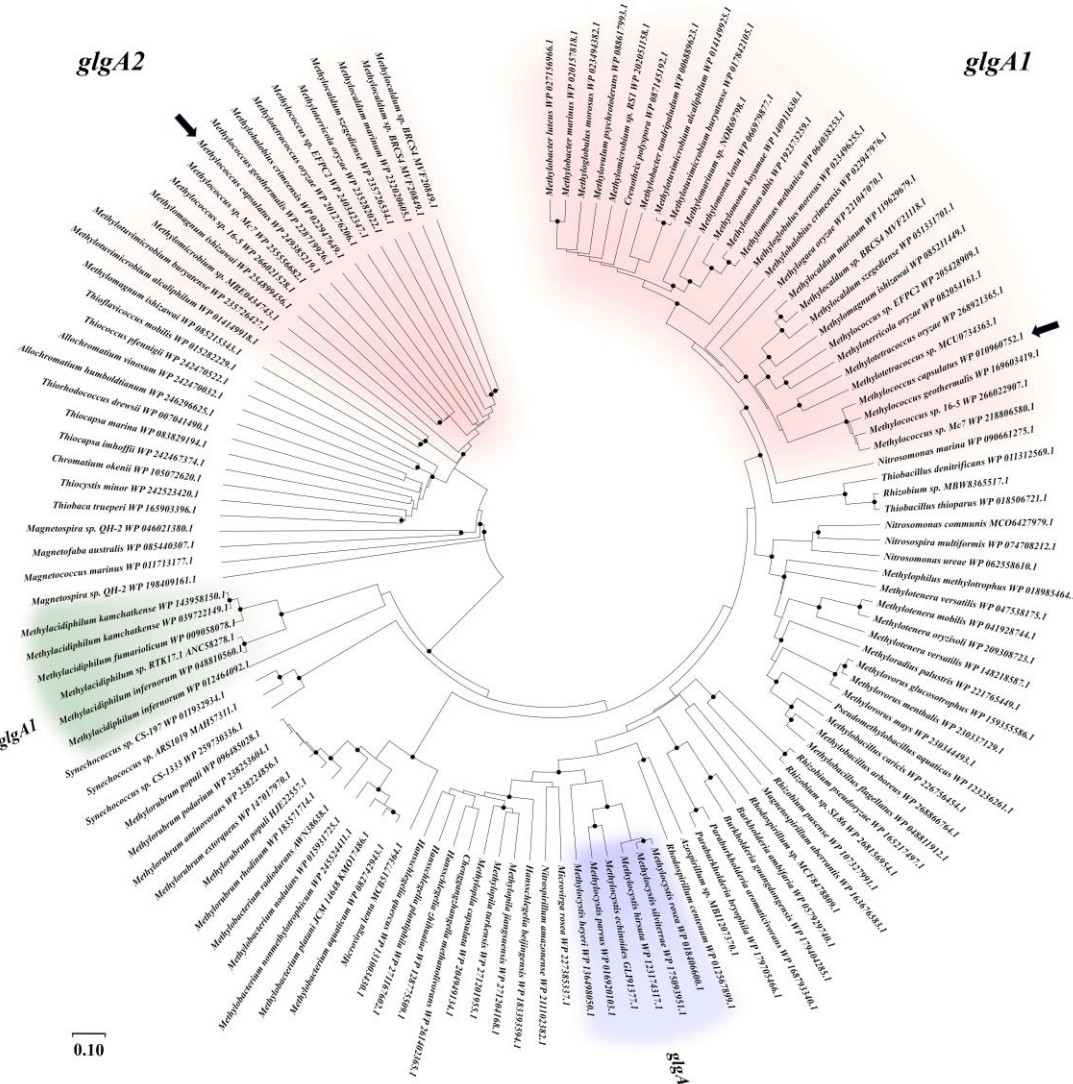

**Figure 1.** Unrooted phylogenetic tree of the glycogen synthase-encoding genes in aerobic methanotrophic bacteria. Methanotrophs of the *Gammaproteobacteria*, *Alphaproteobacteria*, and *Verrucomicrobia* are highlighted in red, blue, and green, respectively. Black arrows point to *glgA1* and *glgA2* genes from *Mc. capsulatus* MIR. Bootstrap values of >80% are shown. Marker, 0.1 substitutions per nucleotide position.

In *Mc. capsulatus* MIR, the *glgA1* gene is located in a cluster containing the *glgB* gene, encoding the putative 1,4-α-glucan branching protein (GlgB), and the *glgC* gene, encoding putative ADP-glucose pyrophosphorylase (GlgC, EC 2.7.7.27). GlgC catalyzes the formation of glycogen precursor ADP-glucose from glucose-1-phosphate and ATP, releasing PPi as another product. This cluster includes the *malQ* gene encoding 4-α-glucanotransferase (MalQ, EC 2.4.1.25) (Figure 2), which is known to preferentially remove free glucose from the reducing end of maltose (or small maltodextrins). The cluster also contains *aspP* gene, coding for the homologue of ADP-sugar pyrophosphatase (AspP, E.C. 3.6.1.21), known to catalyze the hydrolytic breakdown of ADP-glucose to AMP and glucose-1-phosphate [23,39,40]. In *Mc. capsulatus* MIR, the *glgA2*-like gene clusters together with *amy* gene that encodes α-amylase (Figure 2).

**Figure 2.** Scheme of gene clusters encoding glycogen synthesis in *Mc. capsulatus* MIR. *Asp*, gene for ADP-sugar pyrophosphatase; *glgA1*, *glgA2*, glycogen synthase genes; *glgB*, 1,4--α-glucan branching protein; *glgC*, ADP-glucopyrophosphorylase; malQ, gene for MalQ protein (4-α-glucantransferase); *amy*, α-amylase gene.

### 3.2. Construction of Mutant Strains and Phenotypic Characterization

Deletion mutagenesis was successfully applied to obtain Δ*glgA1* and Δ*glgA2* single mutants, as well as Δ*glgA1*Δ*glgA2* double mutant. Previously, the chromosomal deletion of *glk* genes in *Mm. alcaliphilum* 20Z resulted in a significant reduction in glycogen content [41]. Therefore, the glucokinase-deficient strain of *Mc. capsulatus* MIR was also generated.

In the early stationary phase, cells of wild-type (WT) strain MIR grown in the standard medium (with an initial $KNO_3$ concentration of 1 g/L) contained approximately 1.5% glycogen of DCW (Table 2). Under nitrogen-limited conditions, the WT strain accumulated an order of magnitude more glycogen compared to the standard conditions while its protein content decreased. On the contrary, the biomass of Δ*glgA1*Δ*glgA2* strain demonstrated similar protein content regardless of the growth conditions (Table 2).

**Table 2.** Glycogen, glucose, and total protein contents in cells of WT and mutant strains of *Mc. capsulatus* MIR recorded during shake flask cultivation. nd – not determined.

| Growth Conditions | Strain | Glycogen, mg/g DCW | Glucose, μg/g DCW | Protein, mg/g DCW |
|---|---|---|---|---|
| Nitrogen excess (1 g L$^{-1}$ KNO$_3$) | WT | 14.3 ± 3.8 | 32 ± 3 | 496 ± 15 |
| | Δ*glgA1* | 9.4 ± 2.5 | 26 ± 3 | 420 ± 22 |
| | Δ*glgA2* | 17.6 ± 3.2 | 28 ± 1 | 420 ± 17 |
| | Δ*glgA1*Δ*glgA2* | 1.1 ± 0.4 | 35 ± 2 | 477 ± 16 |
| | Δ*glk* | 1.3 ± 0.1 | 75 ± 19 | 498 ± 27 |
| Nitrogen limit (0.29 g L$^{-1}$ KNO$_3$) | WT | 202 ± 54 | nd | 297 ± 63 |
| | Δ*glgA1* | 61.0 ± 9 | nd | 510 ± 24 |
| | Δ*glgA2* | 21.0 ± 3 | nd | 511 ± 52 |
| | Δ*glgA1*Δ*glgA2* | 3.0 ± 0.4 | nd | 532 ± 13 |
| | Δ*glk* | 142.0 ± 5 | nd | 482 ± 36 |

The Δ*glgA1* and Δ*glgA2* single mutants accumulated nearly the same amounts of glycogen as compared to WT strain, but exhibited slightly slower growth rates (Table 2; Figure 2). The protein levels in the dry biomass of the two single mutants were ~85% of those in the WT strain, which suggests that both glycogen synthases were functional and interchangeable. These data also imply that a partial defect in glycogen synthesis in the single *glgA* mutants triggers the rearrangement of metabolism, possibly directing carbon to the synthesis of other intermediates. A double gene deletion mutant, Δ*glgA1*Δ*glgA2*, showed a growth rate nearly equal to that of the WT strain. Both Δ*glgA1*Δ*glgA2* and glucokinase-deficient Δ*glk* strain exhibited only negligible levels of glycogen while maintaining protein contents comparable to that in WT strain. However, the Δ*glk* strain displayed a slightly higher growth rate than the WT strain (Figure 3). Intracellular glucose concentration in

Δ*glk* strain cells was two-fold higher than in the WT strain and in the mutants lacking glycogen synthases.

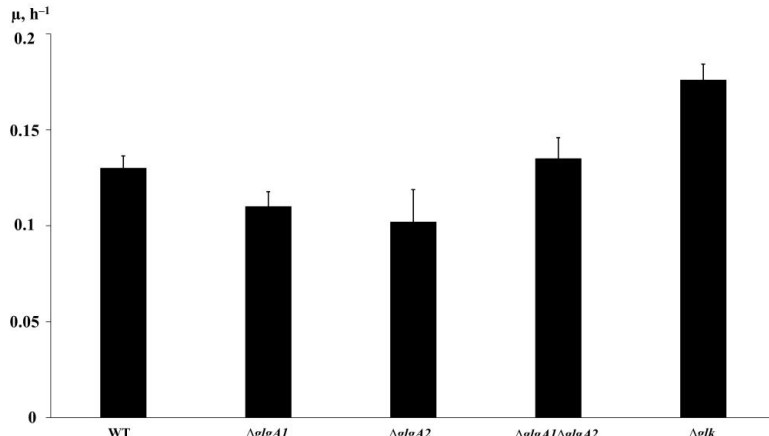

**Figure 3.** Growth rates of WT strain MIR and glycogen-deficient mutants under nitrogen-sufficient conditions (1 g/L KNO$_3$).

### 3.3. Cell Ultrastructure of WT and Mutant Strains under Nitrogen-Limited and Nitrogen-Sufficient Growth Conditions

The cell ultrastructure of the WT strain and the Δ*glgA1glgA2* mutant was examined using transmission electron microscopy. Under nitrogen-sufficient conditions, cells of these strains exhibited similar ultrastructure and contained numerous stacks of intracytoplasmic membranes (Figure 4a,b). Cells of the WT strain grown under nitrogen limitation were filled with numerous glycogen granules (Figure 4c). By contrast, no glycogen granules were observed in electron micrographs of thin sections prepared with cells of strain Δ*glgA1*Δ*glgA2* grown under nitrogen-limited conditions (Figure 4d).

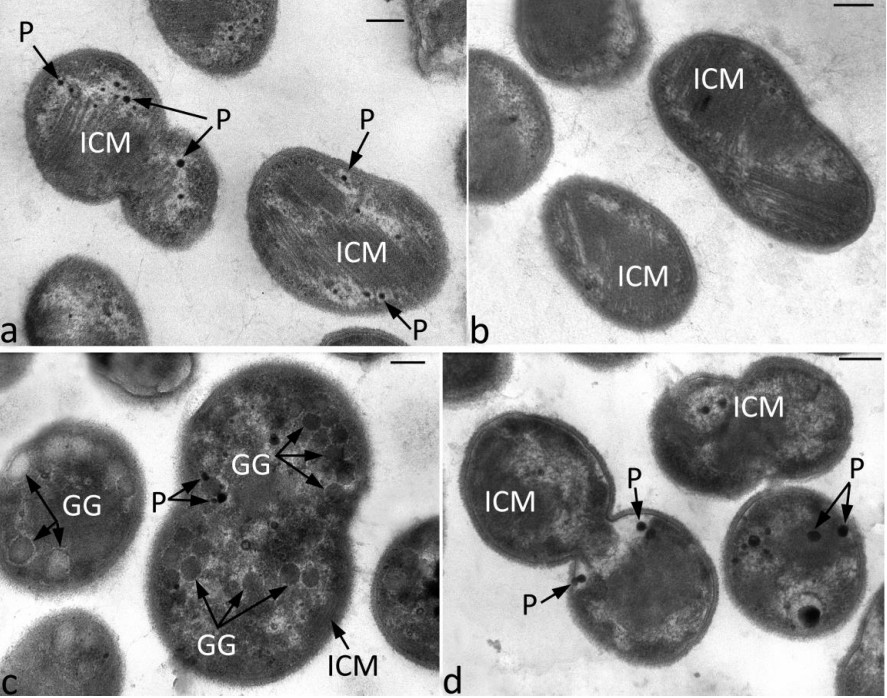

**Figure 4.** Electron micrographs of thin cell sections of *Mc. capsulatus* MIR (**a**,**c**) and Δ*glg1*Δ*glg2* mutant (**b**,**d**). Cells were grown in minimal medium in exponential growth phase with 1 g/L KNO$_3$ (1, 2) or 0.29 g/L KNO$_3$ (3, 4). GG, glycogen granules; ICM, intracytoplasmic membranes; P, inclusions of polyphosphates. Thin sections were stained with uranyl acetate/lead citrate. Bars, 200 nm.

*3.4. Real-Time PCR Analysis of the Genes Responsible for Glycogen Synthesis*

The observed decrease in glycogen content in Δ*glk* mutant cells could be attributed to a drop in the expression of genes directly involved in the polymer synthesis (i.e., glycogen synthases) or genes whose products supply precursors (e.g., ADP-glucose pyrophosphorylase, GlgC; phosphoglucomutase, Pgm). Real-time PCR was employed to assess the expression of these genes. However, experimental results did not confirm this assumption, as the expression levels of *glgA1*, *glgA2*, *glgC*, and *pgm* genes showed no significant variation between the WT strain MIR and the Δ*glk* mutant (Figure 5). The transcript level of AspP, which catalyzes the hydrolysis of ADP-glucose to AMP and glucose-1-phosphate, thus preventing glycogen biosynthesis, was not enhanced. In *E. coli* cells, the activity of AspP was inversely correlated with the intracellular glycogen content and the glucose concentration in the culture medium [23].

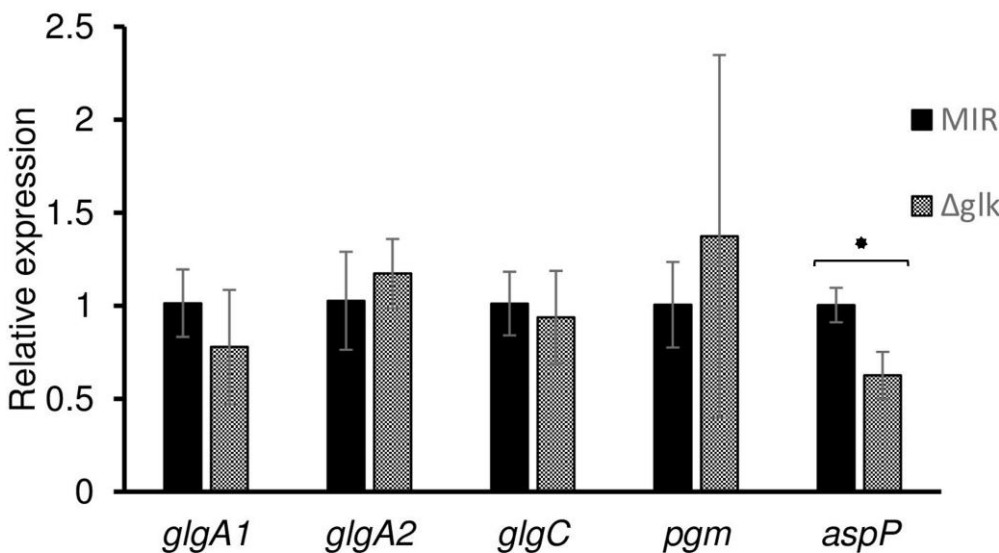

**Figure 5.** Real-time PCR analysis of gene expression in WT strain MIR and glucokinase-deficient mutant strain (Δ*glk*). Examined genes included *glgA1* (glycogen synthase 1), *glgA2* (glycogen synthase 2), *glgC* (ADP-glucose pyrophosphorylase), *pg.* (phosphoglucomutase) and *aspP* (ADP-sugar pyrophosphatase). The *rpoB* gene was used as a reference gene. Three independent experiments were carried out. Bars represent standard deviation. Result with a *p*-value < 0.05 is marked by an asterisk (*).

*3.5. Growth Experiments in a Bioreactor*

Growth of the WT, Δ*glgA1glgA2*, and Δ*glk* strains was also examined in a bioreactor operated in batch and continuous modes (Figure 6).

During batch cultivation, the specific growth rate of WT strain MIR was 0.27 h$^{-1}$, while this parameter reached 0.29 h$^{-1}$ for the Δ*glgA1*Δ*glgA2* and Δ*glk* mutants. Notably, the lag phase was nearly twice as long for the WT strain. The highest biomass yield observed in batch cultures of the WT, Δ*glgA1*Δ*glgA2*, and Δ*glk* strains constituted 4.12, 4.72, and 3.15 g DCW L$^{-1}$, respectively. During continuous cultivation, all strains achieved an almost identical biomass concentration of 3.4–3.5 g DCW L$^{-1}$ at a dilution rate of 0.23–0.25 h$^{-1}$. Culture aliquots were sampled at the early stationary phase before switching to continuous mode and at the steady state of continuous growth. Protein content in the dry biomass of the WT strain increased from 54% to 71% upon reaching steady-state conditions in a continuous culture. Under the same operating conditions, the protein content remained constant at 71% during both batch and continuous growth of strain Δ*glgA1*Δ*glgA2*. In batch and continuous cultures, cells of the Δ*glk* strain contained 66% and 63% of protein, respectively. Specific uptake rates of N by the batch cultures of mutant strains were 61.5 (Δ*glgA1*Δ*glgA2*) and 40.0 (Δ*glk*) mg/g DCW, while the corresponding value determined for WT strain was 52.0 mg/g DCW. Batch culture of WT accumulated a large amount of

glycogen (187.5 mg/g DCW) compared to 10.8 mg/g DCW in Δ*glgA1*Δ*glgA2* cells and 60.4 mg/g DCW in Δ*glk* cells. Glycogen content in the biomass of continuously grown WT, Δ*glgA1*Δ*glgA2*, and Δ*glk* strains constituted 1.6, 0.6, and 4.6 mg/g DCW, respectively.

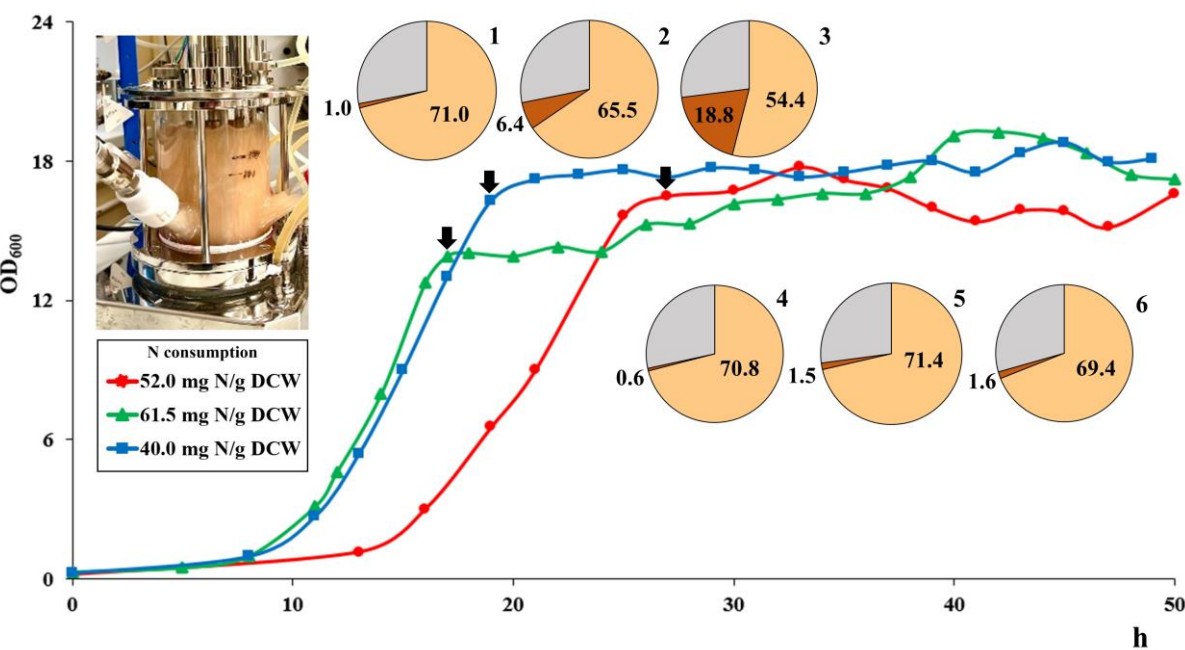

**Figure 6.** Growth dynamics of WT strain MIR (red line), Δ*glgA1*Δ*glgA2* mutant (green line), and Δ*glk* mutant (blue line) during growth in a bioreactor operated in batch and continuous modes. Growth curves represent three independent runs. Transition points from batch to continuous phase are indicated by black arrows. Growth was monitored by measuring OD$_{600}$ values every 2 h. Pie charts show the contents of protein (beige), glycogen (brown), and other cellular components (grey) in dry biomass of Δ*glgA1*Δ*glgA2* (1, 4), Δ*glk* (2, 5), and WT (3, 6) strains during batch (1, 2, 3) and continuous (4, 5, 6) growth. Nitrogen consumption per g DCW for each of the three strains is shown in the frame.

## 4. Discussion

As concluded in specialized assessment studies, SCP produced from methane has a high potential as a resilient food source for global catastrophic food shocks [2]. The product would be affordable at an expected retail cost between US$3–5/kg dry. The technology of SCP production requires three main inputs: (1) methane, which acts as both a carbon source and an electron donor, (2) a nitrogen source, and (3) an oxygen source. Additionally, some minerals are also needed in smaller quantities. As concluded in the analysis by Garcia-Martinez and co-authors [2], most or all of the methane required to fulfill the global protein requirements could be sourced exclusively from a combination of biogas and natural gas associated with oil which is currently being flared or reinjected. Other types of natural gas reserves can also be exploited. Ammonia is used as the source of nitrogen. Notably, an imbalance of C and N availability leads to the accumulation of glycogen in methanotrophic cells which, in turn, results in a loss of protein content and affects the biomass quality as a feed source. A low oxygen-to-methane ratio in the bioreactor can also result in excess carbon flow, potentially leading to glycogen accumulation in the cells. In such cases, employing glycogen-deficient mutants may serve as a strategy to uphold the high quality of SCP by preventing losses in both biomass production rates and protein content.

In this study, thermotolerant methanotroph *Mc. capsulatus* MIR synthesized glycogen when cultivated in a low-nitrogen medium, which aligns with its role as a carbon and energy storage compound. Even the biomass produced by this methanotroph under nitrogen-sufficient conditions contained small amounts of glycogen (~1.5% of dry cell weight) (Table 2). Apparently, due to the low glycogen content, blocking its synthesis did

not have an impact on protein synthesis in strain Δ*glgA1*Δ*glgA2* under nitrogen excess. In addition, a restructuring of carbohydrate metabolism cannot be ruled out, specifically the redirection of carbon flow towards the synthesis of other sugars in response to the knockout of the *glgA* genes. Some examples of such alterations in carbohydrate metabolism have been described previously. Thus, glycogen synthase null mutant obtained from *Synechococcus* sp. PCC7002 synthesized no glycogen but instead produced more soluble sugars excreted into the growth medium via membrane vesicles [42]. *Mc. capsulatus* VSB-874 synthesized large amounts of glycogen (27% of DCW) when cultivated under nutrient excess conditions and excreted exopolysaccharide consisting mostly of glucose and galactose during growth at oxygen or nitrogen limitation [43]. Intracellular polyglucose synthesis was hypothesized as an additional mechanism for formaldehyde binding [43].

These data suggest that glycogen in *Methylococcus* bacteria may serve both as an intermediate metabolite and a long-term storage compound. Recent studies demonstrated that glycogen can be simultaneously synthesized and degraded during bacterial growth [44]. In bioreactor experiments involving WT and Δ*glgA1*Δ*glgA2* strains, a profound difference in glycogen accumulation was observed only during batch cultivation. Glycogen content in biomass of continuously grown Δ*glgA1*Δ*glgA2* mutant was lower only by 1%, which correlated with a slight increase in protein content compared to the WT strain. The mutant strain required more nitrogen for growth during the batch phase, suggesting an increased demand for protein synthesis. Interestingly, the WT strain exhibited a longer lag phase than the glycogen-deficient strain during growth in the bioreactor. This was also observed in flask cultivations. The shorter lag phase demonstrated by glycogen-deficient strain offers a reduction in production cycle time, which is particularly beneficial for bioprocesses utilizing batch, fed-batch, and fill-and-draw cultivation modes for SCP production from methane or natural gas.

Genome analysis showed that the glycogen biosynthesis pathway in *Mc. capsulatus* MIR is generally similar to that in heterotrophic bacteria (Figure 7). The first product of formaldehyde assimilation in the RuMP cycle, fructose-6-phosphate, is transformed into glycogen via glucose-6-phosphate, glucose-1-phosphate, and ADP-glucose as intermediates. This conversion proceeds via sequential reactions catalyzed by hexose phosphate isomerase (Hpi), phosphoglucomutase (Pgm), ADP-glucose pyrophosphorylase (GlgC), glycogen synthases (GlgA1 and GlgA2), and branching enzyme (GlgB). Glycogen catabolism can be controlled by glycogen phosphorylase (GlgP) and debranching enzyme (GlgX).

Further metabolism of glycogen molecules involves α-amylase (Amy) and 4-α-gluca notransferase (MalQ). MalQ is known for its role in removing free glucose from the reducing ends of maltose (or small maltodextrins) and transferring the remaining enzyme-bound dextrinyl residue to other maltodextrins [45,46]. Thus, the breakdown of glycogen results in the formation of free glucose. Glucokinase (Glk) activates glucose by converting it to glucose-6-phosphate that can enter the central metabolic pathways or the new glycogen biosynthesis cycle. Glycogen elongation from glucose-6-phosphate, along with the recapture of this phosphosugar with the participation of Glk, may represent a cycle where two molecules of ATP are transformed into two molecules of ADP and one molecule of PPi.

ADP-sugar pyrophosphatase (AspP) hydrolyzes ADP-glucose to AMP and glucose-1-phosphate, thus preventing glycogen synthesis. The simultaneous activity of GlgC and AspP can create a futile cycle that results in the dissipation of energy stored in ATP. The futile ATP consumption seems to be a reasonable strategy to adapt to nitrogen-limited environments by many bacteria [47,48].

As revealed in our study, the genes encoding glycogen biosynthesis are widely distributed among methanotrophs (Figure 1). Phylogenetic analysis demonstrated that the *glgA* genes from evolutionarily distant methanotrophs formed separate clades, suggesting the vertical inheritance of these genes. Gammaproteobacterial methanotrophs, as well as methanotrophic representatives of the *Verrucomicrobia* possess *glgA1* genes encoding a bacterial-type glycogen synthase GlgA1. The additional (starch-type) glycogen synthase,

GlgA2, is found in some gammaproteobacterial methanotrophs, predominantly in those with halo- and thermotolerant phenotypes (Figure 1).

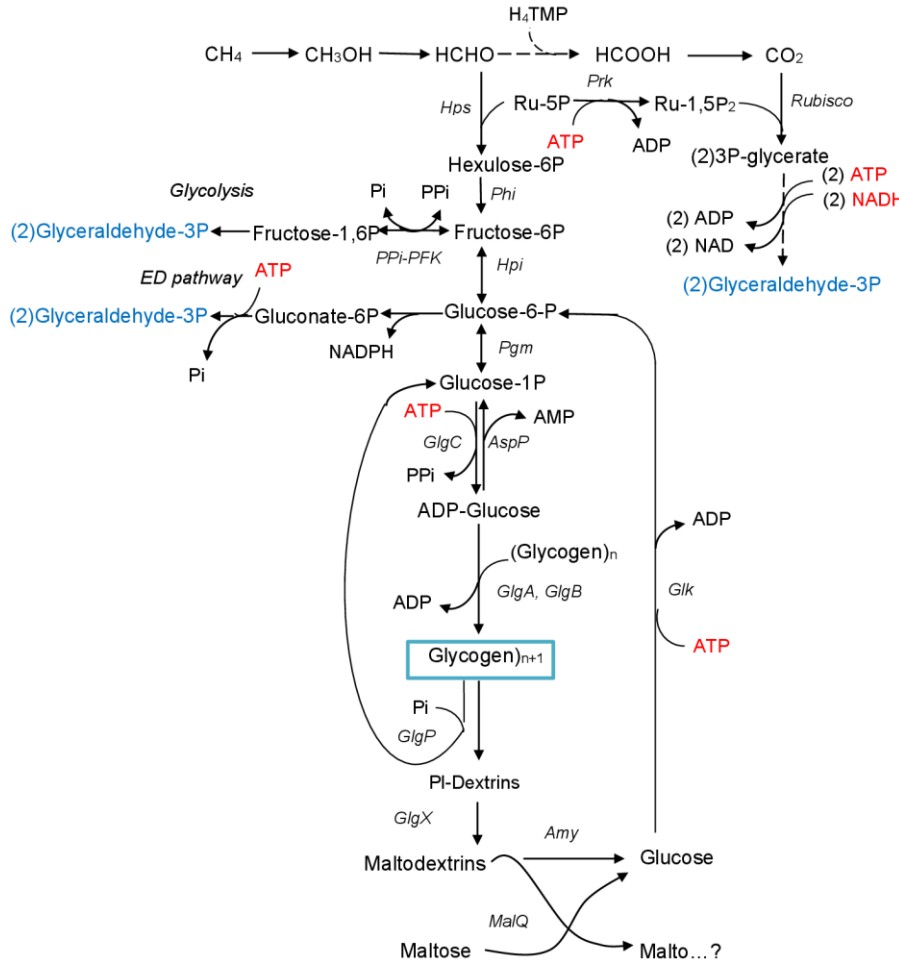

**Figure 7.** Schematic pathways of glycogen metabolism in *Mc. capsulatus* MIR. *Pgm*, phosphogluco-mutase; *GlgA*, glycogen synthase; *GlgB*, 4-α-glucan branching protein; *AspP*, ADP-sugar pyrophos-phatase; *GlgP*, glycogen phosphorylase; *GlgC*, ADP glucose pyrophosphorylase; *GlgX*, debranching enzyme; *MalQ*, 4-alpha-glucanotransferase; *Hps*, hexulosephosphate synthase; *Phi*, phosphohexulose isomerase; *Hpi*, hexosephosphate isomerase; *Prk*, phosphoribulokinase; *Rubisco*, ribulosebisphosphate carboxylase/oxygenase; *PPi-PFK*, pyrophosphate dependent phosphofructokinase.

Inhibition of glycogen synthesis in *Mc. capsulatus* MIR was achieved by simultaneous inactivation of both glycogen synthases. An intriguing finding of our studies was the inhibition of glycogen synthesis by mutating glucokinase in two methanotrophs (see also [41]). *Mm. alcaliphilum* strain Δ*glk* exhibited slower growth compared to the WT strain, while *Mc. capsulatus* strain Δ*glk* grew faster than the WT strain. We believe that the main function of GIK is the removal of intracellular glucose formed during the degradation of glycogen through the *mal* system. Free glucose may have an inhibitory effect on the process of glycogen degradation. In *E. coli*, glucose at concentrations above 0.1 mM was found to block the activity of amylomaltase (MalQ), an enzyme essential for the breakdown of glycogen [49]. In addition, all *mal* genes are thought to be under the positive control of the activator protein MalT [49]. It has been shown that in *E. coli* glucokinase per se is a regulatory protein, forming a complex with MalT. The latter becomes inactive with respect to the initiation of transcription of *mal* genes. Thus, we do not exclude that free glucose may act as a negative regulator of activity or expression of enzymes involved in glycogen biosynthesis and degradation in the methanotroph. However, quantitative real-time PCR

did not reveal significant changes in the transcription levels of the *glgA1*, *glgA2*, *glgC*, *pgm*, or *aspP* genes in cells of the WT and Δ*glk* mutant of *Mc. capsulatus* MIR. Therefore, further studies are required to analyze the effect of glucose on the properties of enzymes involved in glycogen biosynthesis. Differences in the pathways of carbohydrate metabolism may specify different responses upon *glk* elimination in two methanotrophs. In addition to glucokinase, NAD-glucose dehydrogenase (GDH) participates in removing free glucose in cells of strain 20Z, whereas *Mc. capsulatus* MIR lacks genes for GDH [33]. The halotolerant *Mm. alcaliphilum* 20Z accumulated more glycogen than strain MIR [33]. Obviously, this reflects the distinct nitrogen requirements of these bacteria given that *Mm. alcaliphilum* 20Z synthesizes the nitrogen-containing osmoprotector ectoine to maintain the water balance of the cytoplasm [18].

In summary, we demonstrated that the construction of glycogen-deficient mutants of *Methylococcus* species opens new avenues for the production of single-cell protein from methane. Single deletions of *glgA* genes did not significantly impact growth rate and glycogen content in *Mc. capsulatus* MIR, thus suggesting functionality and interchangeability of these two GlgA isoenzymes. Simultaneous inactivation of both *glgA* genes or disruption of the sole *glk* gene resulted in a glycogen-deficient phenotype. Cell biomass of Δ*glgA1*Δ*glgA2* mutant obtained during batch cultivation in a bioreactor displayed high protein content (71% DCW compared to 54% DCW in wild-type strain) as well as a strong (18-fold) reduction in glycogen content. The difference in protein and glycogen contents in biomass of these strains produced during continuous cultivation was less pronounced, yet biomass characteristics relevant to SCP production were slightly better for Δ*glgA1*Δ*glgA2* mutant. Further experiments under controlled conditions are necessary to assess the potential for large-scale applications of these modified methanotrophic strains.

**Author Contributions:** Conceptualization, S.Y.B. and O.N.R.; resources, I.I.M. and N.V.P.; investigation, S.Y.B., O.N.R., R.Z.S. and I.Y.O.; data curation, S.Y.B. and O.N.R.; writing—original draft preparation, V.N.K., S.Y.B., O.N.R. and I.Y.O.; writing—review and editing, V.N.K. and S.N.D.; funding acquisition, S.N.D. and N.V.P. All authors have read and agreed to the published version of the manuscript.

**Funding:** This study was supported by a grant for the Development of genomic editing technologies for innovation in industrial biotechnology (grant no. 075-15-2021-1071) from the Ministry of Science and Higher Education of the Russian Federation.

**Institutional Review Board Statement:** Not applicable.

**Informed Consent Statement:** Not applicable.

**Data Availability Statement:** The raw data and material of the present study are available upon request.

**Acknowledgments:** The authors thank Natalia E. Suzina for the help with electron microscopy studies.

**Conflicts of Interest:** The authors declare no conflict of interest.

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
