# Peer review of "New Solutions in Single-Cell Protein Production from Methane: Construction of Glycogen-Deficient Mutants of Methylococcus capsulatus MIR"

_fermentation, doi:10.3390/fermentation10050265_

Round 1
Reviewer 1 Report
Comments and Suggestions for Authors
In this study, glycogen deficient strain of Methylococcus capsulatus MIR was constructed and analyzed for single cell protein production. In batch cultivation higher protein contents (71% of DCW) were obtained as compared to Wild type strain (54 % of DCW). However, in shake flask experiments and in continuous bioreactor no significant difference in protein contents was observed, despite of minimal glycogen formation. Discussion section is more focused on the glycogen production pathway rather than SCP. This manuscript needs to be revised for publication.
Major comments
1. In table 2, no data have been shown for ΔGIK, ΔglgA1 and ΔglgA2 under nitrogen deficient conditions.
2. Under nitrogen excess condition, despite of glycogen reduction, ΔglgA1ΔglgA2 double mutant strain has almost similar levels of protein contents as compared to WT strain. Does it imply that glycogen formation has no effect on protein contents? Please discuss this result.
3. GIK catalyzes glucose-6-phosphate production, which is an important intermediate in cellular metabolism. Therefore, what are the possible reasons for increased growth rate of ΔGIK strain? GIK deficient strain of M. alcaliphilum 20Z showed similar behavior for glycogen formation in a previous study. Authors can refer to this publication for discussion (DOI: 10.1007/s10482-016-0809-z)
4. ΔGIK strain showed higher protein contents comparing with ΔglgA1ΔglgA2, still this strain was neither tested under nitrogen deficient conditions nor in bioreactor.
5. Figure 6 should include the nitrogen consumption in bioreactor.
Minor comments:
1. Line 14: fast growing thermotolerant aerobic methanotroph? Please revise the abstract.
2. Line 173: Please provide the reference.
3. Line 234: “The GlgA1 from strain MIR and the corresponding enzyme from E. coli display 46% sequence identity”. Author is referring to which strain of E. coli here?
4. In figure 6 what is the grey color in circle?
5. Line 352: Change the word “unlimited” to excess.
6. Gene abbreviations should be italic in text and figures.
7. Is strain abbreviation Mc. Capsulatus and Mm. alcaliphilum is correct? Please revise in the whole manuscript.
8. Line “252” does not seem complete. Please revise.
9. Table 3: please provide the references
Comments on the Quality of English LanguageNo problem.
Author Response
Comment: Discussion section is more focused on the glycogen production pathway rather than SCP. This manuscript needs to be revised for publication.
Response: We have added some additional text on SCP production in the Discussion and also reduced the text on glycogen production pathway.
Major comments:
Comment: In table 2, no data have been shown for ΔGIK, ΔglgA1 and ΔglgA2 under nitrogen deficient conditions.
Response: We have performed additional experiments and obtained these data, which are now included in table 2.
Comment: Under nitrogen excess condition, despite of glycogen reduction, ΔglgA1ΔglgA2 double mutant strain has almost similar levels of protein contents as compared to WT strain. Does it imply that glycogen formation has no effect on protein contents? Please discuss this result.
Response: Since only a low amount of glycogen (~1.5% of dry cell weight) was accumulated in cells of WT strain growing under nitrogen excess conditions, reduction of glycogen content in cells of ΔglgA1ΔglgA2 mutant strain did not make a noticeable contribution to the overall biomass composition. Some restructuring of carbohydrate metabolism cannot be ruled out as well, in particular, a redirection of carbon flow to the synthesis of other (phospho)sugars in response to the removal of glgA genes. Several examples of such alterations in carbohydrate metabolism were earlier described in the literature. Thus, a decrease in intracellular polyglucose content in cells of Mc. capsulatus 874 in response to oxygen limitation was accompanied by the accumulation of a large amount of extracellular heteropolysaccharide consisting of glucose and galactose in a ratio of 1:14 (Khmelenina et al., 1992 Ref. 41). Another example is glycogen synthase null mutant of Synechococcus sp. PCC7002, which synthesized no glycogen but produced more soluble sugars excreted into growth medium via membrane vesicles (Xu et al., 2012).
Comment: GIK catalyzes glucose-6-phosphate production, which is an important intermediate in cellular metabolism. Therefore, what are the possible reasons for increased growth rate of ΔGIK strain? GIK deficient strain of M. alcaliphilum 20Z showed similar behavior for glycogen formation in a previous study. Authors can refer to this publication for discussion (DOI: 10.1007/s10482-016-0809-z)
Response: We believe that the main function of GIK is the removal of intracellular glucose formed during the degradation of glycogen through the mal system. Perhaps free glucose has an inhibitory effect on the process of glycogen synthesis and degradation in methanotrophs. In E. coli, glucose in concentrations above 0.1 mM blocked the activity of amylomaltase (MalQ), an enzyme required for the breakdown of glycogen (Lengsfeld et al., 2009). In addition, all mal genes are thought to be under the positive control of the activator protein MalT (Richet, Raibaud, 1989). It has been shown that in E. coli glucokinase per se is a regulatory protein, forming a complex with MalT, the latter becomes inactive with respect of initiation of transcription of mal genes. We believe that one of the reasons for the accelerated growth of the glucokinase-deficient mutant is the cessation of the synthesis of certain enzymes of glycogen metabolism.
In a previous study we found that GlK from strain 20Z had biochemical properties similar to other prokaryotic GlKs. The glk deficient strain of Mm. alcaliphilum 20Z demonstrated behavior similar to that of Mc. capsulatus MIR with regard to glycogen formation (Mustakhimov et al., 2017, ref 40; Rozova et al., 2021). The data obtained suggested that GlK is implicated in the regulation of glycogen biosynthesis/degradation in an obligate methanotroph. Glucose-6-phosphate is indeed an important intermediate product of cellular metabolism, being also a precursor to glycogen. Since glucose-6-phosphate is formed at an early stage of C1-substrate assimilation through the ribulose monophosphate cycle, inhibition of the synthesis and breakdown of glycogen with the formation of this phosphosugar is a more efficient process of its biosynthesis. Mm. alcaliphilum 20Z has additional enzyme for glucose metabolism i.e., glucose dehydrogenase catalyzing NAD+-dependent conversion of glucose in gluconic acid.
Comment: ΔGIK strain showed higher protein contents comparing with ΔglgA1ΔglgA2, still this strain was neither tested under nitrogen deficient conditions nor in bioreactor.
Response: We have performed additional experiments and obtained these data, which are now included in table 2 and are shown in Figure 6.
Comment: Figure 6 should include the nitrogen consumption in bioreactor.
Response: These data are included in this figure now.
Minor comments:
Comment: Line 14: fast growing thermotolerant aerobic methanotroph? Please revise the abstract.
Response: Done.
Comment: Line 173: Please provide the reference.
Response: Done.
Comment: Line 234: “The GlgA1 from strain MIR and the corresponding enzyme from E. coli display 46% sequence identity”. Author is referring to which strain of E. coli here?
Response: This refers to the E. coli K-12. The corresponding corrections have been made.
Comment: In figure 6 what is the grey color in circle?
Response: These are other cellular components. We clarify this now in the figure caption.
Comment: Line 352: Change the word “unlimited” to excess.
Response: Corrected as recommended.
Comment: Gene abbreviations should be italic in text and figures.
Response: Done.
Comment: Is strain abbreviation Mc. capsulatus and Mm. alcaliphilum is correct? Please revise in the whole manuscript.
Response: Yes, these abbreviations are correct. the corresponding edits have been made throughout the manuscript.
Comment: Line “252” does not seem complete. Please revise.
Response: Done.
Comment: Table 3: please provide the references
Response: This table is sort of redundant since it shows essentially the same info as in Figure 1. This table, therefore, has been omitted from the revised manuscript.
Reviewer 2 Report
Comments and Suggestions for Authors
In this study, the authors explored a new solutions in single-cell protein production from methane by constructing a glycogen-deficient mutants of Methylococcus. The production of single-cell proteins through bacterial cells has certain significance, and the author has achieved some positive results. Therefore, I suggest accepting the manuscript after appropriate revisions.
1. Methane itself is an energy source, does converting methane into protein have economic value?
2. In line 157, for centrifugation process, the author needs to use g instead of rpm, and detailed information about the centrifuge also needs to be presented.
3. For reagents or instruments, the author needs to provide detailed information (model, company, city, and country).
4. For the Figure 5, whether gene expression in WT strain MIR and glucokinase-deficient mutant strain (Δglk) have significant differences?
5. For the Figure 6, the author's presentation is confusing. Did different strains of bacteria be added to the same reactor?
6. For GLK, it has the fastest growth rate. Which one in Figure 6 corresponds to this strain?
Author Response
Comment: Methane itself is an energy source, does converting methane into protein have economic value?
Response: The technology of converting methane into protein has clear economic value in countries with big gas reserves, such as Russia, USA, Norway and others. Apart from economically exploitable reserves, there are stranded gas reserves, which cannot currently be economically exploited for typical industrial uses. One particular example is gas associated with oil reserves and thus requiring extraction before the oil can be exploited. This gas is often regarded as an undesirable byproduct of oil extraction and is commonly flared, reinjected, or vented. Analysis of the economic potential of producing methanotrophic microbial protein from stranded methane produced at wastewater treatment plants, landfills, and oil and gas facilities is given by Garcia Martinez et al. (ref 2). As stated by these authors, methane SCP would be affordable at an expected retail cost between US$ 3-5/kg dry weight.
Comment: In line 157, for centrifugation process, the author needs to use g instead of rpm, and detailed information about the centrifuge also needs to be presented.
Response: Done.
Comment: For reagents or instruments, the author needs to provide detailed information (model, company, city, and country).
Response: The corresponding information has been added in the revised manuscript.
Comment: For the Figure 5, whether gene expression in WT strain MIR and glucokinase-deficient mutant strain (Δglk) have significant differences?
Response: We have replaced Figure 5 with a new figure showing a statistically significant difference (p<0.05).
Comment: For the Figure 6, the author's presentation is confusing. Did different strains of bacteria be added to the same reactor?
Response: Growth curves represent three independent runs and were included in one figure for comparison purposes. This is now explained in the figure caption.
Comment: For GLK, it has the fastest growth rate. Which one in Figure 6 corresponds to this strain?
Response: There was no Δglk growth curve in the previous version of this figure. We have performed additional growth experiments in a bioreactor and obtained the corresponding growth curve for Δglk strain. It is now included in revised Figure 6.
Round 2
Reviewer 1 Report
Comments and Suggestions for Authors
In accordance with reviewers' comments, the authors have successfully addressed the comments and revised manuscript well. The revised version of manuscript can be accepted as is for publication.
Reviewer 2 Report
Comments and Suggestions for Authors
The author has fully considered my suggestion and made corresponding modifications, and I suggest accepting it in the current version.